# Beneficial Effects of Jujube Juice Fermented by *Lactobacillus plantarum* NXU19009 on Acute Alcoholic Liver Injury in Mice

Huiyan Liu [1], Shihua Xin [1,2], Ranran Lu [1], Haitian Fang [1,*], Xiaoping Yang [2] and Yun Ping Neo [3]

[1] Ningxia Key Laboratory for Food Microbial-Applications Technology and Safety Control, School of Food & Wine, Ningxia University, Yinchuan 750021, China; Liuhy@nxu.edu.cn (H.L.); nanshan0526@gmail.com (S.X.); sq809364@gmail.com (R.L.)

[2] School of Tourism Management, Ningxia Vocational College of Industry and Commerce, Yinchuan 750021, China; sq877153092@gmail.com

[3] School of Biosciences, Taylor's University Lakeside Campus, Subang Jaya 47500, Malaysia; yunping.neo@taylors.edu.my

* Correspondence: fanght@nxu.edu.cn

**Abstract:** Red jujube (*Ziziphus jujuba* Mill.) is an important fruit that has the concomitant function of both medicine and food. It has been proven to be rich in various bioactive components. In the present study, jujube juice was fermented by *Lactobacillus plantarum* NXU19009 to enhance the flavor and nutritional benefits. Its potential for the prevention and treatment of acute alcohol induced-liver injury in mice was examined in this study. The results showed that the administration of the fermented jujube juice along with alcohol significantly decreased ($p < 0.01$) the liver indices, as well as the levels of alanine aminotransferase (ALT), aspartate aminotransferase (AST), alcohol dehydrogenase (ADH), total triglyceride (TG), total cholesterol (TC), and liver malondialdehyde (MDA) in the serum. In contrast, the levels of liver superoxide dismutase (SOD) and glutathione (GSH) in mice administered with fermented jujube juice were found to increase significantly ($p < 0.01$). Furthermore, the administration of fermented jujube juice in mice was found to alter their intestinal microbiota and an improvement was observed based on the results obtained in the histopathology examination. Therefore, Jujube juice fermented by *Lactobacillus plantarum* NXU19009 protects against liver injury and may prove to be an effective supplement to attenuate acute alcoholic liver injury.

**Keywords:** jujube juice; *Lactobacillus plantarum* NXU19009; fermentation; acute alcoholic liver injury; intestinal microbiota



## 1. Introduction

Red jujube (*Ziziphus jujuba* Mill) is an indigenous fruit widely cultivated in northern China that serves as both food and medicine. It is known for its rich nutritional value and functional components, such as organic acids [1], polysaccharides, flavonoids [2], and triterpenoids, that were reported to exhibit anti-inflammatory and antibacterial properties [3–5]. Another important functional component of red jujube is the cyclic adenosine phosphate (cAMP), which is an important regulator in human physiological processes, especially to improve liver function [6,7]. In addition, there are also studies on the liver protection of diketones and triterpenoids from the jujube [2,7]. As a result, jujube has garnered much interest regarding its potential to treat or prevent alcoholic liver disease, whereby research on the application, antioxidant activity, and hepatoprotective effect of jujube have been reported [1]. To date, most reports have focused on the jujube polysaccharides, triterpenes, and cAMP, and research related to the hepatoprotective effect of fermented jujube juice is scarce, despite lactic acid bacteria (LAB) having also been found to ameliorate liver injury [8].

It is worth mentioning that the LAB strain as a starter culture for fermentation has been found to enhance the nutritional properties of food [9,10]. Studies have reported the

capability of LAB fermentation to improve the antioxidant properties of plant-based food. Furthermore, LAB could also contribute to gut microbiota and regulate the disorder of gut microbes by inhibiting the reproduction of pathogenic bacteria [11,12]. Many in vivo and in vitro studies also have proven the efficacy of LAB to regulate the intestinal microbiota, promote intestinal peristalsis, reduce serum cholesterol, and inhibit the growth or reproduction of intestinal spoilage bacteria [13]. Some studies have also suggested that LAB could protect the gastric mucosa injury and improve the microenvironment of the gut [8].

Acute drinking can lead to abnormal changes of physiological metabolism, such as rapid glycogen consumption, hypoglycemia, acidosis, and acute alcoholic liver disease (ALD) [14]. In recent years, the incidence of ALD has increased substantially [15]. More evidence indicates that ROS-mediated oxidative stress plays a crucial role in the pathogenesis of ALD [16]. ROS generated by the Cytochrome P450 2E1 (CYP2E1) enzyme was found to affect lipid metabolism such as the index of triglycerides (TG), low-density lipoprotein (LDL), and oxidative stress, such as glutathione (GSH) and malondialdehyde (MDA) [17]. As a result, more studies started to look into the effects of gut microbiota in regulating liver disease, as intestinal bacteria may alter the expression of P450 enzymes. Furthermore, the imbalance of the intestinal microbiota has also been proven to be a key factor in the occurrence and development of ALD [18]. Regulating and maintaining stable, together with diverse intestinal microbiota, to a certain extent, might provide a potential target for the prevention and treatment of ALD.

In the present study, jujube juice was fermented by *Lactobacillus plantarum* NXU19009, which is expected to enhance the production of bioactive metabolites that will protect the liver. The alleviation effects of lactic-acid-fermented jujube juice towards ALD was investigated and verified using an acute alcohol-induced liver injury model in mice. The serum biochemical indices, liver oxidation index, liver inflammatory factor expressions, liver pathology, and the intestinal microbiota of the mice were determined. The findings of the present study provide a better understanding of the preventive mechanism of fermented jujube juice on ALD, which can serve as useful information for the commercial applications of jujube fruit and fermented jujube juice.

## 2. Materials and Methods

### 2.1. Jujube Juice Fermentation

Dried red jujube fruits (from Ningxia, *Ziziphus jujuba* Mill.) were washed, peeled, stoned, and mixed with water (1:6 g/mL) in a homogenizer. Then, 0.15% sodium ascorbate and pectinase were added to the jujube pulps, which were then dissolved and digested for 2 h at 50 °C in a water bath. The residues of jujube were removed by centrifugation, and the juices were sterilized at 80 °C for 30 min in a water bath. After sterilization, the jujube juices were cooled to 40 °C. The initial pH of the jujube juices was 6.5 and the initial °Brix was 19.4 before fermentation. Next, 6% (*v/v*) *L. plantarum* NXU19009 were allotted and inoculated into the sterilized jujube juices at 37 °C for 18 h. The viable cell counts of the *L. plantarum* as an inoculum in jujube juice at the beginning of the fermentation was 4.8 logCFU/mL, and was found to be approximately 8.41 logCFU/mL after 18 h of fermentation. Raw jujube juices with no inoculation were cultured and used as the control. All the fermentations were carried out in triplicate.

The *L. plantarum* NXU19009 strain used in the present study was supplied by Ningxia Key Laboratory for Food Microbial-Applications Technology and Safety Control. The bacteria culture was stored at −80 °C in an MRS medium with 16% glycerol. The strains were reactivated after inoculating 6% (*v/v*) of the culture in 100 mL of sterile MRS broth and incubating at 37 °C for 24 h. After the second transfer in MRS broth, the activated culture was used for the fermentation.

### 2.2. Animals and Experimental Design

Specific pathogen free (SPF) male Kunming mice weighing 23–27 g were provided by the animal experiment center of Tianjin University of Science and Technology, and

were housed under controlled temperature (23 ± 2 °C) with a 12-h light/dark cycle. Free drinking and eating were provided to the mice. After 7 days of adaptive feeding, the mice were randomly divided into six groups (*n* = 10 per group), including the normal group (N), which was the control group administered 10 mL/kg/d purified water; the model group (M), which was the alcoholic liver injury model group administered with 10 mL/kg/d purified water; the positive group (P), which was the alcoholic liver injury model group administered with 15 mL/kg/d purified water containing 0.01% liver-protecting medicine (Neptune Jinzun, Wanbangde Pharmaceutical Group Co., Ltd., Wenling, China); the fermented low-dose group (L), which was the alcoholic liver injury model group administered with 10 mL/kg/d of the fermented jujube juice; the fermented high-dose group (H), which was the alcoholic liver injury model group administered with 20 mL/kg/d of the fermented jujube juice; and the unfermented group (U), which was the alcoholic liver injury model group administered with 10 mL/kg/d of unfermented raw jujube juice. Four hours after the last gastric administration, mice from N were administered intragastrically with physiological saline 10 mL/kg/d, and mice from the other groups were administered with alcohol 10 mL/kg/d (53° Red Star Erguotou, Beijing Red Star Co., Ltd., Beijing, China) for 30 days. The mice from each group were fasted for 12 h and were sacrificed by cervical dislocation. They were weighed and the collected feces were immediately placed in liquid nitrogen and subsequently transferred to a −80 °C freezer. The blood samples of all experimental groups were collected and centrifuged at 3500 r-min for 10 min to separate the serum. The liver tissue samples were weighed, marked, and stored in the −80 °C freezer.

### 2.3. Measurement of Serum Biochemical Indices and Liver Index

The levels of TG, total cholesterol (TC), aspartate aminotransferase (AST), alanine aminotransferase (ALT), and alcohol dehydrogenase (ADH) in the serum samples were measured according to the instructions of the corresponding assay kits (Nanjing Jiancheng Bioengineering Institute, Nanjing, China). The mice liver index was calculated based on the following formula: (mice liver wet weight/body weight) × 100%.

### 2.4. Histopathology Analysis and Hepatic Antioxidant Enzymes Assay

A small portion of liver tissue was mixed with a nine-fold amount of phosphate buffer solution (PBS) in an ice bath to prepare a 10% tissue homogenate. The homogenate was centrifuged at 12,000 rpm at 4 °C for 15 min, and the supernatant was used for MDA, GSH, and superoxide dismutase (SOD) assays, according to the corresponding kits (Nanjing Jiancheng Bioengineering Institute, China). The pathology of the liver was analyzed using the tissue section method. Briefly, the liver tissue was fixed in 10% neutral buffered formalin for 24 h. Paraffin blocks embedded with the liver tissues were cut into 5 μm thick sections and stained with Hematoxylin−Eosin (HE), as previously reported [19]. The sections were observed under a light microscope (Olympus, Shinjuku, Japan).

### 2.5. Analysis of Stool Intestinal Microbiota in Mice

The fecal samples from each experimental group (*n* = 6) were collected and extracted using a DNA Extraction Kit (Takara, Tokyo, Japan). The amplicon library was constructed by amplifying the V3-V4 hypervariable region of the 16S rDNA gene using the universal primer (338F-ACTCCTACGGGAGGCAGCAG, 806R-GGACTACHVGGGTWTCTAAT), and was sequenced by the Beijing Genomics Institute (Shenzhen) Co., Ltd. The PCR amplification was carried out using the following conditions: initiation at 95 °C for 5 min starting 20 cycles at 95 °C for 30 s, 55° C for 30 s, and 72 °C for 30 s with a final extension at 72 °C for 10 min. The amplicons were purified and dissolved in an elution buffer. The amplicon libraries were established and sequencing was conducted utilizing the HiSeq platform.

*2.6. Statistical Analysis*

The measurement data were expressed as mean ± standard error. Data were analyzed using SPSS software (version 19.0, SPSS, Chicago, IL, USA) and one-way ANOVA followed by Origin 2017 software (OriginLab, Austin, TX, USA). Post hoc tests were performed for data comparisons among groups. Differences were statistically significant when $p < 0.05$.

## 3. Results

*3.1. Growth Status and Liver Index of Experimental Mice*

The weight change and liver index of the mice were determined to analyze the beneficial effects of the fermented jujube juice after intragastric administration. As shown in Table 1, the weight gain of the M group mice was found to be significantly lower ($p < 0.05$) than for the N group. Similarly, M group mice also showed a significantly higher ($p < 0.05$) liver index compared to the other groups. The average body weight of all of the experimental mice groups increased by 26.70%, 15.71%, 7.28%, 1.94%, 8.44%, and 4.45% for the N, M, L, H, U, and P groups, respectively (Table 1), compared to their initial body weight. The average body weights of the P group and the H group were found to be significantly lower ($p < 0.05$) than for the rest. Overall, the alcohol-induced increase in the liver index was found to be significantly lower ($p < 0.05$) in groups treated with fermented and unfermented jujube juice.

**Table 1.** Effects of the fermented jujube juice on the weight and liver index in mice with acute alcoholic liver injury.

| Groups | Initial Body Mass/g | Final Body Mass/g | Liver Index/(g/100 g) |
|---|---|---|---|
| N | 29.74 ± 1.28 | 37.68 ± 2.29 | 3.86 ± 0.13 |
| M | 33.74 ± 4.29 | 39.04 ± 2.91 | 4.09 ± 0.23 ## |
| L | 33.09 ± 1.52 | 35.5 ± 2.99 | 3.76 ± 0.20 * |
| H | 34.04 ± 1.65 | 34.7 ± 2.49 | 3.57 ± 0.15 * |
| U | 33.16 ± 1.71 | 35.96 ± 3.47 | 3.49 ± 0.21 |
| P | 33.05 ± 3.34 | 34.52 ± 3.75 | 4.04 ± 0.18 ** |

N—normal group; M—model group; H—fermented jujube juice high-dose group; L—fermented jujube juice low-dose group; U—unfermented jujube juice group; P—positive group (liver-protecting medicine). $n = 10$. Statistical significance is marked as * represents the significant difference between each experimental group and the N group with $p < 0.05$, ** represents $p < 0.01$; ## represents the significant difference between each experimental group and the M group with $p < 0.01$. The data were the means ± standard error of at least three independent experiments. All of the comparisons were performed with Origin 2017 software (OriginLab, USA), utilizing the unpaired t-test with Welch's correction.

*3.2. Serum TG, TC, AST, ALT and ADH Activities*

The levels of TG, TC, AST, ALT, and ADH in the mice serum are shown in Table 2 and Figure 1. The TG, TC, AST, ALT, and ADH levels of the M group mice serum were significantly higher ($p < 0.01$) compared to the N group, indicating a successful establishment of an alcohol-induced liver injury model in the present study. The levels of serum TG, TC, AST, ALT, and ADH generally showed a significant reduction ($p < 0.01$) for the L, H, U, and P groups compared to the M group. Interestingly, the L and H group exhibited lower levels of serum AST and ALT compared to the U group.

*3.3. Hepatic GSH, SOD, and MDA Levels*

Several indicators of liver oxidative stress, including GSH, SOD, and MDA concentrations, were measured in the present study to determine the effects of fermented jujube juice on the antioxidant capacity in the mice livers (Table 3 and Figure 2). Table 3 shows the GSH and SOD activities of the M group were significantly reduced ($p < 0.01$) by 47.05% and 35.33%, respectively, compared to the N group, whereas the MDA content of the M group was found to be significantly increased ($p < 0.01$) by 52.2% compared to the N group, indicating the impairment of the antioxidant capacity due to acute alcoholism.

**Table 2.** Effects of the fermented jujube juice on serum biochemical indices in mice with acute alcoholic liver injury.

| Groups | AST (IU/L) | ALT (IU/L) | ADH (U/mL) | TG (mmol/L) | TC (mmol/L) |
|---|---|---|---|---|---|
| N | 8.22 ± 0.74 | 7.31 ± 0.90 | 1.66 ± 0.13 | 1.27 ± 0.21 | 3.23 ± 0.13 |
| M | 12.83 ± 0.17 ** | 12.50 ± 1.02 ** | 2.02 ± 0.07 ** | 2.61 ± 0.27 ** | 6.28 ± 0.19 ** |
| L | 10.49 ± 0.49 **## | 9.91 ± 1.23 **## | 1.84 ± 0.04 ** | 1.70 ± 0.17 *## | 4.57 ± 0.16 **## |
| H | 9.86 ± 0.83 **## | 9.83 ± 1.25 **## | 1.95 ± 0.03 *## | 1.73 ± 0.22 ## | 4.47 ± 0.24 **## |
| U | 10.89 ± 0.47 **## | 10.13 ± 1.68 **## | 1.97 ± 0.04 ** | 2.31 ± 0.38 * | 4.89 ± 0.20 **## |
| P | 8.89 ± 0.39 **## | 9.55 ± 0.32 **## | 1.87 ± 0.03 *## | 1.65 ± 0.34 ## | 4.14 ± 0.17 **## |

N—normal group; M—model group; H—fermented jujube juice high-dose group; L—fermented jujube juice low-dose group; U—unfermented jujube juice group; P—positive group (liver-protecting medicine). $n = 10$. Statistical significance is marked as follows: * represents the significant difference between each experimental group and the N group with $p < 0.05$, ** represents $p < 0.01$; ## represents the significant difference between each experimental group and the M group with $p < 0.01$. The data were the means ± standard error of at least three independent experiments.

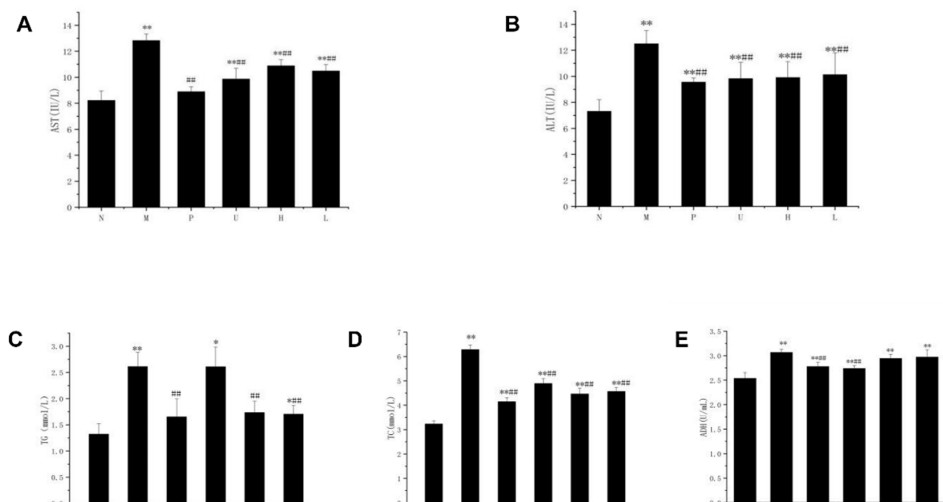

**Figure 1.** Comparative analysis of the serum biochemical indices in mice with acute alcoholic liver injury. (**A**) serum AST levels; (**B**) serum ALT levels; (**C**) serum TG levels; (**D**) serum TC levels; (**E**) serum ADH levels. Mice were divided into the following categories: N—normal group; M—model group; P—positive group; U—unfermented group; H—fermented high-dose group; L—fermented low-dose group. $n = 10$. Statistical significance is marked as follows: * represents the significant difference between each experimental group and the N group with $p < 0.05$, ** represents $p < 0.01$; ## represents the significant difference between each experimental group and the M group with $p < 0.01$.

**Table 3.** Effects of the fermented jujube juice on hepatic SOD, GSH, and MDA in mice with acute alcoholic liver injury.

| Groups | GSH (mmol/L) | SOD (U/mg) | MDA (mmol/L) |
|---|---|---|---|
| N | 175.18 ± 7.87 | 99.12 ± 11.44 | 2.56 ± 1.09 |
| M | 92.75 ± 25.35 ** | 64.10 ± 4.85 ** | 5.47 ± 0.62 ** |
| L | 119.13 ± 18.95 **# | 85.28 ± 7.36 *## | 4.78 ± 1.17 ** |
| H | 160.74 ± 23.65 ## | 74.98 ± 4.87 **# | 3.72 ± 1.16 *## |
| U | 132.71 ± 12.79 **## | 66.57 ± 2.39 ** | 3.83 ± 0.91 |
| P | 165.50 ± 11.92 ## | 89.25 ± 6.30 *## | 3.37 ± 0.75 ## |

N—normal group; M—model group; H—fermented jujube juice high-dose group; L—fermented jujube juice low-dose group; U—unfermented jujube juice group; P—positive group (liver-protecting medicine). $n = 10$. Statistical significance is marked as follows: * represents the significant difference between each experimental group and the N group with $p < 0.05$, ** represents $p < 0.01$; # represents the significant difference between each experimental group and the M group with $p < 0.05$, ## represents $p < 0.01$. The data are the means ± standard error of at least three independent experiments.

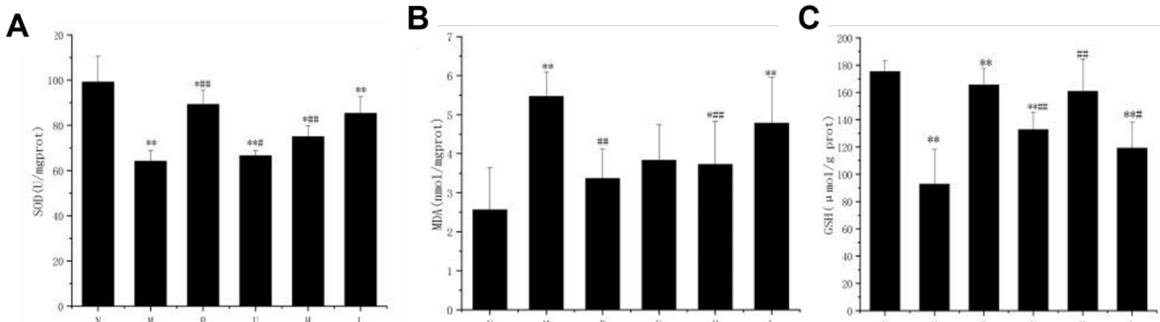

**Figure 2.** Effects of the fermented jujube juice on hepatic (**A**) SOD, (**B**) MDA, and (**C**) GSH activities. Mice were divided into the following categories: N—normal group; M—model group; P—positive group; U—unfermented group; H—fermented high-dose group; L—fermented low-dose group. $n = 10$. Statistical significance is marked as follows: * represents the significant difference between each experimental group and the N group with $p < 0.05$, ** represents $p < 0.01$; # represents the significant difference between each experimental group and the M group with $p < 0.05$, ## represents $p < 0.01$.

Figure 2A,C shows that the L, H, U, and P groups generally exhibited significantly higher hepatic antioxidant activities compared to the M and N groups. Similarly, Figure 2B shows that the MDA contents of the H and P groups were significantly lower ($p < 0.01$) than the M group.

### 3.4. Histopathological Analysis

The liver histopathological changes in the mice were assessed after HE staining under a microscope (×200 times). As shown in Figure 3A, the structure of the normal liver tissue (N group) was complete, the hepatocyte cords were arranged neatly, the cytoplasm was rich, and the liver cell tissue morphology was normal. Compared with the N group, the liver cell boundaries were unclear, hepatocyte cords were disorderly arranged, the spacing became larger and swollen, and the hepatocytes had moderate edema and showed spotted necrosis in the M group (Figure 3B). The P group exhibited complete liver tissue structure and some hepatocytes showed microvesicular steatosis (Figure 3C). As shown in Figure 3D,E the liver tissue structures were complete, but slight steatosis was observed in the L and H groups. Figure 3F reveals microvesicular steatosis in the livers of the U group (Figure 3F).

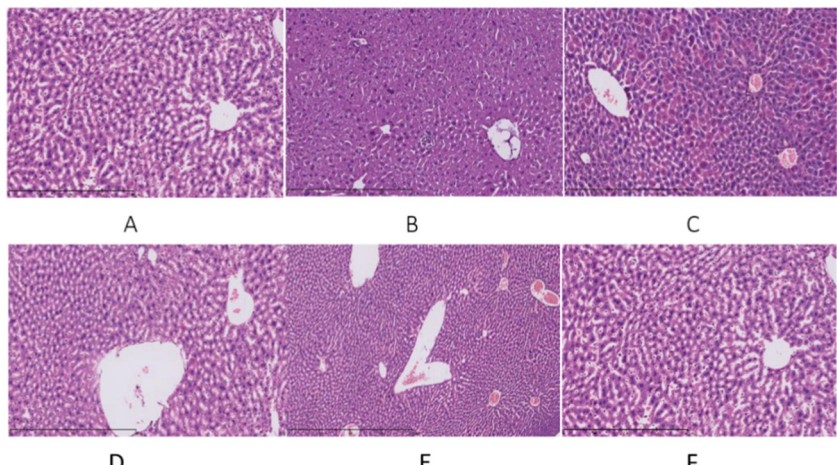

**Figure 3.** Observation of liver pathology in mice with acute alcoholic liver injury. (**A**) N, the normal group; (**B**) M, the model group; (**C**) P, the positive group (liver-protecting medicine); (**D**) L, the fermented jujube juice low-dose group; (**E**) H, the fermented jujube juice high-dose group; (**F**) U, the unfermented jujube juice group.

*3.5. Effects of the Fermented Jujube Juice on the Mice Intestinal Microbiota*

Shannon-Wiener is an index that reflects the health diversity in the sample. The larger the value, the higher the community diversity [20]. As shown in Figure 4A, the average of the Shannon-Wiener index values of the H group was significantly higher than for the other groups, and the number of bacterial groups were significantly increased ($p < 0.01$) compared to others. Figure 4B shows a total of 14 genus bacteria were detected in the present study, mainly the *Bacilliformis, Akkermansia, Bacteroides,* and *Escherichia*. The limit abundance and proportion of intestinal microbiota pacing in each experimental group were as follows: *Lactobacillus* (20.61%), *Bacteroides* (8.43%), and *Acinetobacter* (3.86%) were the most prominent in the N group; *Pediococcus* (18.72%), *Lactobacillus* (10.27%), *Bacteroides* (7.01%), *Enterococcus* (6.59%), and *Escherichia* (5.24%) were the highest in the M group; *Akkermansia* (24.46%), *Lactobacillus* (18.98%), *Enterococcus* (11.52%), *Acinetobacter* (4.63%), and *Clostridium* (4.00%) were the dominant species in the L group; *Lactobacillus* (20.16%), *Bacteroides* (10.68%), *Prevotella* (9.00%), and *Akkermansia* (8.20%) were the most prominent in the H group; *Pediococcus* (25.27%), *Bacteroides* (18.15%), *Akkermansia* (15.00%), and *Prevotella* (4.31%) were the highest species in the U group; *Lactobacillus* (12.34%), *Akkermansia* (11.65%), *Bacteroides* (9.94%), *Enterococcus* (8.21%), and *Prevotella* (7.69%) were the dominant species in the P group.

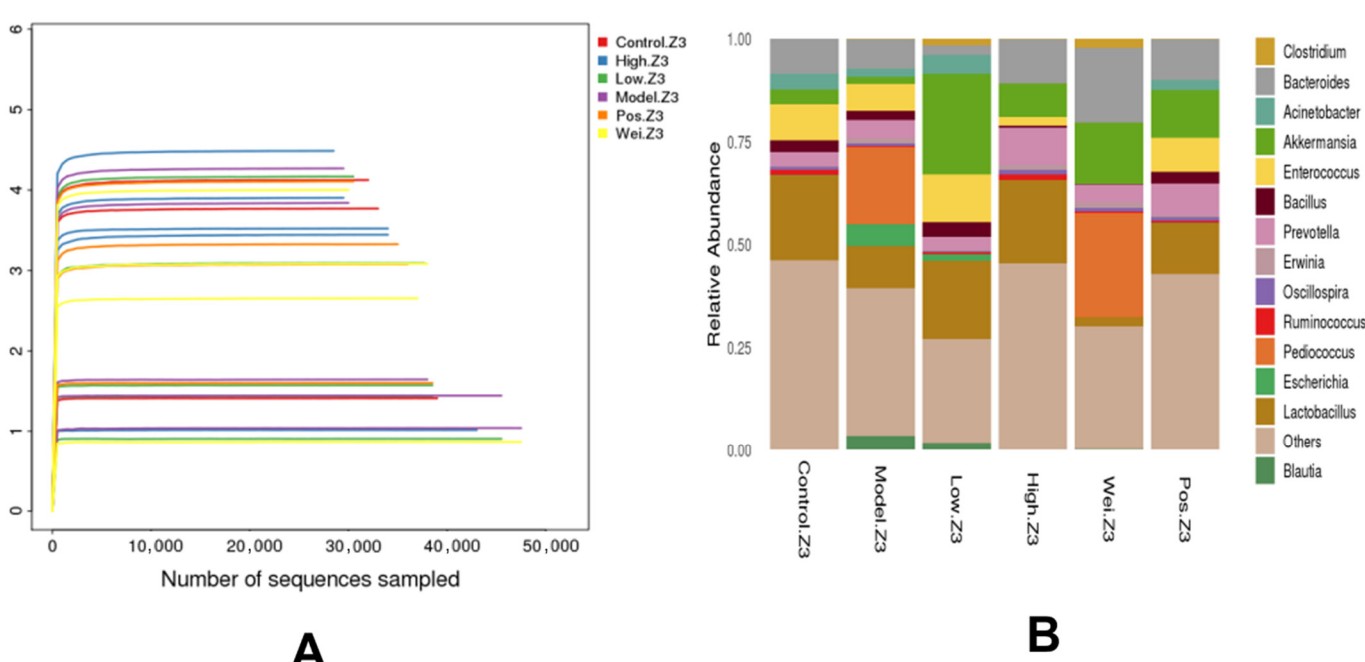

**Figure 4.** (**A**) The Shannon-Wiener index curve. (**B**) The horizontal abundance map of the intestinal microbiota in mice.

## 4. Discussion

The degree of liver injury in mice was reflected through the liver indices in this study. Acute alcoholic injury would lead to a decrease of the mouse body weight together with an increase in the liver index. Both fermented and unfermented jujube juice reduced the mice body weight after intragastric administration. The results obtained in this study implied the potential effects of the fermented jujube juice in attenuating liver injury by preventing the incidence of alcohol-induced liver enlargement.

The effects of fermented and unfermented jujube juice on the levels of TG, TC, ALT, AST, and ADH in the serum were evaluated in the present study. Studies have shown that an excessive intake of alcohol can cause a significant increase in serum ALT and AST levels [21], characterized by the increased hepatocyte membrane permeability or hepatocellular necrosis. ALT and AST are the two most important enzymes distributed

in the liver cytoplasm and mitochondria. The activities of these two enzymes usually serve as indicators of liver cell damage [22]. Under normal circumstances, the activities of these two enzymes in the serum are low. Nevertheless, it has been suggested that these enzymes will be released into the blood slowly after hepatocellular injury [23]. Hence, determination of the activities of these two enzymes in the serum is adequate to serve as a prognosis for acute liver injury. Studies have found that excessive drinking can also affect the lipid metabolism [24]. Excessive intake of alcohol can cause lipid metabolism disorders, which lead to the accumulation of fat in the liver [25]. When fat changes occur in the liver, the blood lipid concentration will also increase [26]. As a result, TG and TC contents in the serum are good measures of lipogenesis in a cell to reflect the pathology of dyslipidemia [27]. The levels of ALT, AST, TG, TC, and ADH in the M group with acute alcoholic liver injury were higher than those of the N group, while the L, H, U, and P group displayed significantly reduced activities in these biochemical indices. Based on the results obtained, the significant reduction of these biochemical indices in the serum therefore indicated that fermented jujube juice is able to regulate the lipid metabolism and prevent liver steatosis. Similarly, ADH in the liver is one of the important enzymes in the body to metabolize ethanol into acetaldehyde (toxic metabolite of alcohol) and further oxidizes the acetaldehyde to acetic acid. A higher ADH activity may lead to increased alcohol metabolism and a reduced risk of alcohol induced liver damage [28,29]. The administration of fermented jujube juice increases the ADH activity in mice as compared to the normal group, indicating its potential to accelerate the metabolism of alcohol and aldehyde levels in the blood, thereby displaying a protective effect on alcoholism.

High levels of alcohol consumption can result in the formation of ROS in our body, whereby oxidative stress plays an important role in the development of alcoholic liver disease. Excessive drinking may reduce the levels of SOD and GSH (endogeneous enzymatic antioxidant), and may increase the level of MDA (lipid peroxidation indicator) in our bodies. The administration of fermented jujube juice increased the levels of SOD and GSH and decreased the levels of MDA in the present study, which implies its effect to inhibit lipid peroxidation by strengthening the endogeneous antioxidant defense system. Furthermore, the hepatic morphology of the mice examined in this study showed that the administration of fermented jujube juice improved the histophatological changes of the liver tissues by alleviating their steatosis state, which corroborated its protective effect on acute alcoholic liver injury.

The intestinal microbiota plays a vital role in the development of ALD. Various studies have shown the effect of alcohol consumption on the intestinal microbiome composition, which can lead to malnutrition due to suboptimal absorption of various nutrients [30]. The mice group administered with fermented jujube juice generally exhibited increments in the abundance of intestinal *Bacteroides* and *Lactobacillus* compared to the M group, resulting in a higher level of beneficial microbes. The specific taxonomical identification implied the prospects of fermented jujube juice to modulate our intestinal microbiota in the prevention and treatment of alcohol-induced liver injury.

## 5. Conclusions

In summary, an acute alcohol-induced liver injury model was established in the present study. Based on the results obtained from the measurements of different biochemical indices, along with histopathological analysis, the jujube juice fermented by *L. plantarum* exhibited a comprehensive effect in alleviating alcoholic induced liver injury and improving the intestinal microbiota. Further works on the inflammatory response and glycolipid metabolism profile of fermented jujube juice will help to better understand its contribution in regulating acute alcoholism and strengthen its role as a functional food.

**Author Contributions:** Conceptualization, H.F.; methodology, H.L.; software, R.L. and S.X.; validation, H.F. and R.L.; formal analysis, H.L.; investigation, S.X. and R.L.; resources, X.Y.; data curation, R.L.; writing—original draft preparation, H.L.; writing—review and editing, Y.P.N.; supervision, project administration and funding acquisition, H.F. All authors have read and agreed to the published version of the manuscript.

**Funding:** This research was funded by the key R&D program of Ningxia (2018BBF02008), the innovation platform funds of Ningxia key laboratory for food microbial-applications technology and safety control (2019BDC05009, 2019YDDF0062), and the project of support local colleges and universities reform and development funds of China (2018).

**Institutional Review Board Statement:** All the experimental procedures followed the "Guidelines for the Care and Use of Laboratory Animals (Eighth Edition, ISBN-10: 0-309-15396-4)". The study was conducted in accordance with the regulations of Chinese Law and approved by the Institutional Animal Ethics Committee of Ningxia University (Animal experiment permission number: SYXK2017-005; Date: 11 October 2019). The animal study protocol was in compliance with the regulations of the Institutional Animal Ethics Committee and was approved by Beijing Viton Lee Huashi Experimental Animal Technology Co., Ltd. (Animal Production License Number: SCXK 2017-0005, Date: 11 October 2019).

**Informed Consent Statement:** Not applicable.

**Data Availability Statement:** Data sharing not applicable.

**Conflicts of Interest:** The authors declare no conflict of interest.

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
