# Peer review of "Beneficial Effects of Jujube Juice Fermented by Lactobacillus plantarum NXU19009 on Acute Alcoholic Liver Injury in Mice"

_fermentation, doi:10.3390/fermentation8020054_

Round 1

Reviewer 1 Report

The manuscript is well written and presents interesting results regarding the effect of fermented fruit juice consumption on liver injury using mice as a model. However, some improvements are needed before publication:

Lines  40-48 – not all lactic acid bactéria are probiotics to be claimed probiotic several requirements must be accomplished. There are some bactéria that are probiotic for amimals but no for humans, their microbiota are diferent. Please see the recent consensus on probiotics and fermented foods available at https://isappscience.org/for-scientists/resources/probiotics/. The manuscript generalizes stating that all LAB are probiotics, and that is not true.

Line 71 – correct to jujube pulp

296 – change the word invention to study

Author Response

Dear Reviewer,

On behalf of my co-authors, we appreciate you very much for your positive and constructive comments and suggestions on our manuscript. We have tried out best to revise the manuscript carefully according to your comments.

The main corrections in the paper and the responds to the editor’s comments are as flowing:

1.Lines  40-48 – not all lactic acid bactéria are probiotics to be claimed probiotic several requirements must be accomplished. There are some bactéria that are probiotic for amimals but no for humans, their microbiota are diferent. Please see the recent consensus on probiotics and fermented foods available at https://isappscience.org/for-scientists/resources/probiotics/. The manuscript generalizes stating that all LAB are probiotics, and that is not true.
Response: Thanks for your kindly direction. I got some information about probiotics from https://isappscience.org/for-scientists/resources/probiotics/ . I agree with you very much, and we have revised some words according to your advice.

2. Line 71 – correct to jujube pulp

Response: Thank you for your advice,the wrong spelling has been corrected and marked in red.

3.296 – change the word invention to study

Response: Thank you very much for your suggestion, inaccurate word has been corrected and marked in red.

We would like to express our great appreciation again to you and look forward to hearing from you.

With best wishes,

Prof. Haitian Fang ([email protected]), Mrs. Huiyan Liu ([email protected]

Reviewer 2 Report

Authors studied the potential beneficial effects of starter driven fermented jujube juice on acute alcoholic liver injury in mice. The work is very interesting and the aim is clear. However, substantial revisions are needed before further consideration.

The Introduction should be condensed.

In discussion section, authors should describe and explain their findings more deeply, as well as in a more suitable way.

An overall conclusion about the usefulness and scientific impact of the work is totally missing. How scientific community could be benefited by the present work? Furthermore, what is recommended as the next step?

In many parts of the text the description is not satisfactory, making the reading very difficult to be followed. Furthermore, there are several language, grammatically and other errors (too numerous to be listed) and thus, a drastic language revision and a general improvement throughout the manuscript are necessary.

-Lines 29-32. Please revise.

-Line 40. It is “lactic acid fermentation juice”.

-Lines 40-43. Please revise.

-Line 54. Please replace the term “flora” to “microbiota” throughout the manuscript.

-Lines 64-66. Please revise.

-Lines 73-74. Please add more information about inoculation conditions (LAB growth, temperature, population load used for inoculation, etc.). What 6% inoculation stands for? This is not enough. Furthermore, which are the properties of the starter (both technological and probiotics). Please specify.

-Lines 107-116. The description of the metabarcoding analysis is not satisfactory described. Please rephrase the whole paragraph following a rational description structure.

-Lines 231-233. What is the meaning of figure 4A? I suggest providing a table instead of figure. Furthermore, in Figure 4B the relative abundances are expressed as percentage.

-Lines 117-250. There are many parts in results section that should be moved to discussion. Please revise carefully.

Author Response

Dear reviewer,

On behalf of my co-authors, we appreciate you very much for your positive and constructive comments and suggestions on our manuscript. We have tried out best to revise the manuscript carefully according to your comments.

The main corrections in the paper and the responds to your comments are as flowing:

1.Authors studied the potential beneficial effects of starter driven fermented jujube juice on acute alcoholic liver injury in mice. The work is very interesting and the aim is clear. However, substantial revisions are needed before further consideration.

Response: Thanks for your kindly direction.We have revised the abstract according to your advices.

2.The Introduction should be condensed.

Response:Thank you for your advice, the Introduction has been condensed and revised in detail in the resubmitted manuscript.

3.In discussion section, authors should describe and explain their findings more deeply, as well as in a more suitable way.

Response: Thanks for your kindly direction. We have revised the discussion section according to your advices.

4.An overall conclusion about the usefulness and scientific impact of the work is totally missing. How scientific community could be benefited by the present work? Furthermore, what is recommended as the next step?

Response: Thanks for your suggestion. We have resummarized the usefulness and scientific impact of the conclusion section to ensure that the scientific community can obtain valuable information.

5.In many parts of the text the description is not satisfactory, making the reading very difficult to be followed. Furthermore, there are several language, grammatically and other errors (too numerous to be listed) and thus, a drastic language revision and a general improvement throughout the manuscript are necessary.

Response: Thank you for your valuable and thoughtful comments. We have carefully checked and improved the English writing in the revised manuscript. So we invited Dr. Yun-Ping Neo from the Taylor’s University of Malaysia to improve the language of the manuscript and list her as a co-author. And revised in detail in the resubmitted manuscript.

  1. Lines 29-32. Please revise.

Response: Thanks.We have revised.

  1. Line 40. It is “lactic acid fermentation juice”.

Response: Thanks.We have revised.

  1. Lines 40-43. Please revise.

Response: Thanks.We have revised.

  1. Line 54. Please replace the term “flora” to “microbiota” throughout the manuscript.

Response: Thanks.We have revised.

10.Lines 64-66. Please revise.

Response: Thanks.We have revised.

  1. Lines 73-74. Please add more information about inoculation conditions (LAB growth, temperature, population load used for inoculation, etc.). What 6% inoculation stands for? This is not enough. Furthermore, which are the properties of the starter (both technological and probiotics). Please specify.

Response: Thanks for your suggestion. We have revised and supplemented in the resubmitted manuscript.

  1. Lines 107-116. The description of the metabarcoding analysis is not satisfactory described. Please rephrase the whole paragraph following a rational description structure.

Response: Thanks for your suggestion. We have revised and supplemented in the resubmitted manuscript,and marked in red.

13.Lines 231-233. What is the meaning of figure 4A? I suggest providing a table instead of figure. Furthermore, in Figure 4B the relative abundances are expressed as percentage.

Response: Thanks. Just to show that the amount of sequencing data is large enough to reflect the vast majority of microbial information in the sample on the Shannon-Wiener index curveof figure 4A. 

14.Lines 117-250. There are many parts in results section that should be moved to discussion. Please revise carefully.

Response: Thanks for your suggestion. We have revised and some results be moved to discussion section in the resubmitted manuscript.

We would like to express our great appreciation again to you and look forward to hearing from you.

With best wishes,

Prof. Haitian Fang ([email protected]),

Mrs. Huiyan Liu ([email protected]

Round 2

Reviewer 2 Report

The authors have appropriately addressed my comments.